# Impact of Atmospheric Polycyclic Aromatic Hydrocarbons (PAHs) of Falling Dust in Urban Area Settings: Status, Chemical Composition, Sources and Potential Human Health Risks

**DOI:** 10.3390/ijerph20021216

**Published:** 2023-01-10

**Authors:** Mohamed Hamza EL-Saeid, Abdulaziz G. Alghamdi, Abdulhakim Jari Alzahrani

**Affiliations:** Department of Soil Sciences, College of Food and Agriculture Sciences, King Saud University, P.O. Box 2460, Riyadh 11451, Saudi Arabia

**Keywords:** human health index, organic pollutant, PAHs, road dust, public health, cancer risk

## Abstract

The present work is considered to investigate the sources, concentration, and composition of polycyclic aromatic hydrocarbons (PAHs) and associated health risk assessment of road dust in Riyadh City, Saudi Arabia. The study region included an urban area, strongly affected by traffic, a bare and an industrial area. A total of 50 locations were selected for sampling and 16 different PAHs were determined. The concentration of PAHs in road dust and their estimated lifetime average daily dose (LADD) for adults (human) ranged from 0.01 to 126 ng g^−1^ and 1950 to 16,010 mg kg^−1^ day^−1^, respectively. The ADDing was calculated separately for children (>6), teenagers (6–12), and adults (>12) for all PAHs with each collected sample. Moreover, the average daily exposure dose by ingestion (ADDing) and average daily exposure dose by dermal absorption (ADDder) were more in children (<6 years) as compared to teenagers (6–12 years) and adults (>12 years). Likewise, total equivalency factor based on BaP (TEQBaP) calculations pointed out that PAHs having more benzene rings or having high molecular weight showed high TEQBaP as compared to low molecular weight PAHs. The data revealed that the children population is at high risk for asthma, respiratory and cardiovascular diseases, and immunity suppression as compared to adults in the particular area of investigated region. These outcomes of this study can be used to deliver significant policy guidelines concerning habitants of the area for possible measures for controlling PAHs contamination in Riyadh City to protect human health and to ensure environmental sustainability.

## 1. Introduction

The PAHs are produced because of partial thermal decomposition and combustion of inorganic sources during industrial and power production activities, and desalination of water [1,2]. Natural sources are also present, such as volcanoes and forest fires [3,4]. Riyadh City, in Saudi Arabia, is a highly densely populated area, with several commercial and trade manufacturing regions having multiple anthropogenic sources of pollutants. During the past two decades, rapid growth because of urban and industrial expansion has occurred in Riyadh City. Nowadays, the urbanized, as well as industrialized regions, covers more than five thousand square kilometers. Population data predict an increase in the population of Riyadh up to 7.2 million by 2023. The main roads in Riyadh City consist of 3982 km, as well as 814 km of intercity ordinary roads [5,6]. The rapid expansion of any area results in dense traffic, diesel combustion, ultimately with outcomes of PAHs, heavy metals, and POPs release into the environment [7,8].

Previous studies indicated that the concentration of PAHs in air of Riyadh City was in the range of 1.8 to 13.5 μg/m^3^. The major PAH sources were characterized by petroleum product emissions caused by automobile combustion products and diesel emissions [9]. Local input is considered to be the main source with a minor influx from longer-range transport [10]. Therefore, it is anticipated that a variety of compounds are released during diesel combustion leading to potential environmental pollution that may negatively affect human health, especially in children [11,12]. In another study, the PAHs BaP, cyclopentaphenanthrene, benzoacephenanthrylene, anthracene, chrysene, benz(a)anthracene, naphthalene, phenanthrene, and fluorine were detected in the blood serum of children suffering from asthmatic allergy [13,14]. revealed that PAH concentrations were found to be highest in the serum of people who tend to live in the Riyadh vicinity, which was prone to heavy traffic and industrial emissions. The deposition rate of road dust in Riyadh City in 2016 was assessed to be about 392 t km^−1^ year^−1^ [6]. This value represents a large increase from an earlier estimation (i.e., 220 t km^−1^ year^−1^) that took place back in 1987 [15]. Road dust adhering to surfaces can lead to significant problems for urban management. Road dust can easily re-suspend in specific outdoor environments, and adsorbed pollutants may enter the bodies of humans and animals through direct skin contact and respiratory inhalation pathways, which poses a threat to health [16]. Aging people are more susceptible to such PAHs toxins since of their weak or age-compromised immune systems [17].

Polycyclic aromatic hydrocarbons (PAHs) are among the toxins in road dust and are the most recognized main pollutant class among all contaminants [18]. PAHs are among the persistent organic pollutants that are carcinogenic and mostly cause skin, gastrointestinal, lung, and bladder cancers [19]. They also possess lipophilic characteristics [20,21,22] have highlighted that PAHs pose serious risks related to carcinogenic health issues. Nitro-PAHs are another class of PAHs that constitute nitro substituents [23]. Like PAHs, Nitro-PAHs are produced through similar anthropogenic sources among which coal combustion, vehicle exhaust, and biomass burning are the main emitting sources [24]. In addition, the secondary reaction of PAHs with nitrogen oxide, ozone, and hydroxyl can result in the formation of nitro-PAHs; however, in the condensed phase, homo and hetero transformations carried out [25]. In broad-spectrum, some PAHs such as nitrated polycyclic hydrocarbons have high concentration as magnitude as related to their concentration in the environment. The toxicity of these PAHs tends to be highest [26,27]. Thus, surface dust is a complex environmental medium that acts as a non-point source of pollution [28]. Research has shown that surface dust comprises inputs from a wide range of sources, such as material from the nearby soils (transported through water), dry and wet atmospheric conditions, particles as a result of paint deterioration, particulate emissions, vehicle fluids, and particles from the weathering of buildings and sidewalks [29]. Assessing the levels of PAHs in road dust can be used to deliver significant guidance concerning possible measures for controlling pollution sources (non-point), improving environmental quality, and protecting human health. Therefore, the study was designed to identify the concentrations of PAHs in road dust collected from different areas in Riyadh City and to calculate the different factors which are associated with human health risks.

## 2. Materials and Methods

### 2.1. Study Site and Geography

Riyadh City is located in the middle of the Arab Peninsula in the heart of the desert at 24.7136° N and 46.6753° E. The altitude of Riyadh City is 600 m, and the city expands on an area of 1800 km^2^. The weather of Riyadh City is hot and dry, with temperatures ranging from 9 °C to 53 °C, and humidity from 10 to 50% throughout the year.

### 2.2. Collection and Preparation of Dust Samples

Falling dust was sampled from Riyadh City (Figure 1) during the summer season. Samples were obtained from 50 different residential locations in the east, west, north, and south along with central city areas. Ten representative samples (covering almost all areas) were collected for PAHs analysis from each area with a vacuuming method using the Royal Vacuum Brand in triplicate. The dust samples were taken from the top surface in each area and sealed in plastic bags, brought back to the Department of Soil Sciences, King Saud University, Riyadh, Saudi Arabia, for all analyses. The samples were prepared for measurements by being sieved through a 100 µm mesh screen to eradicate building debris and macro biological materials, including rodent droppings [30].

### 2.3. Dust Samples Preparation and Extraction by QuEChERS

The special extraction procedure was adopted by taking 2 g of dust sample and adding 1 mL of deionized water in a 50 mL centrifuge tube, vortexing shortly, and after vortexing, allowing it to homogenate for about 10 min total. Acetonitrile (6 mL) was added to each sample. Shaking (5 min) was carried out for the extraction of PAHs. The substances of citrate salts (ECQUEU750CT-MP) Mylar pouch were added to individual dust samples in the centrifuge tube. Instantaneously, samples were shaken for a minimum of 2 min and centrifuged for 5 min at ≥3500 rcf. Sample cleanup was carried out by transferring a 1.5 mL aliquot of supernatant to a 2 mL CUMPSC18CT (MgSO4, PSA, C18) dSPE tube. Samples were vortexed for 2 min and again centrifuged for 2 min at high rcf (i.e., ≥5000). The supernatant solution, using a 0.2 μm syringe filter, was transferred straight away into a GC vial (1.8 mL), while at the end, PAHs in the extracts solutions were determined by GC-MS/MSTQD.

### 2.4. PAHs Analysis by GCMSTQD

The EPA method was used to customize the GC MS/MS conditions (SVOC 8270). The auto SRM was used for the method development of MS/MS. To obtain good sensitivity, the modified method was divided into quantifier and qualifier ions. Scanning was performed in each segment (500 to 700 MS) while the maximum transitions were 51 per segment. Auto tuning of MS was performed before each batch of analysis, while helium (purity: 99.99%) gas was used as a carrier gas, nitrogen (purity: 99.9999%), argon (purity: 99.9999%) was used as collision gas obtained from a registered firm Linde gas (SiGas, Saudi Arabia). A total of sixteen (16) PAHs were analyzed, and their description and allotted symbols are shown in Appendix A.

### 2.5. Risk Assessment of Human Health

The soil absorption pathway was well thought out as the key pathway of PAHs for determining human health risks. The lifetime average daily dose (LADD) of PAHs through the soil is calculated in this study based on the concentration emitted from one box to the next. For the risk assessment analysis, exposure parameters established by the [31] have been used. Related studies in the literature have used these response parameters all over the world. The following formula was used to measure LADD:LADD (mg Kg^−1^day^−1^) = (Cs × IR × F × EF × ED)/(BW × AT)(1)
where Cs is the individual concentration of PAH in soil (μg kg^−1^), IR is the soil ingestion rate, F is a unit factor, EF is the exposure frequency (day year^−1^), ED is the lifetime exposure duration (year), BW is the body weight (70 kg for adult’s calculation), and AT is the average time for carcinogens (day).

### 2.6. Health Risk Index

The health risk index is globally applied to measure surface dust particles’ potential health risks [32,33]. Dermal absorption and hand-to-mouth ingestion have been recognized as the major entrance pathways for surface dust (toxins) to enter the human body. The health risk index includes calculating the following:*ADDing* = C × BA (IngR × TEF × ED)/(BW × AT) × 10^−6^(2)
*ADDder* = C × (SL × SA × ABS × TEF × ED)/(BW × AT) × 10^−6^(3)
where *ADDing* is the daily dose from the hand-to-mouth ingestion of substrate particles, and the *ADDder* is the daily dose via dermal absorption of PAHs in particles stuck to exposed skin. The characteristics of all remaining parameters used in Equations (2) and (3), and their adopted values are presented in Table 1.

### 2.7. Toxicity Equivalents of PAHs in Soils

The toxic equivalency factor (TEF) of BaP (TEQBaP) has been proposed as a capable approximation for determining the toxicology of PAHs in road dust toxic equivalent concentrations [35]. The TEF was designed to determine the danger posed by composite combinations, such as dioxins, which may help discern more precisely the carcinogenic components in composite combinations. The methodology defines a compound’s separable matrix toxic potency, which is typically the deadliest complex in a mixture. In the case of PAHs, BaP is thought to be the most active and has a well-defined toxicological profile [35]. The concentration of PAHs at each sampling site was converted into TEQBaP using the corresponding TEF [35] according to the following equation:TEQBaP = ∑(C × TEF)(4)
where C is the concentration of a PAH.

### 2.8. Statistical Analysis

To compare the mean values and outcomes from all parameters, a descriptive statistical analysis technique was performed using Microsoft Excel^®^ 2016.

### 2.9. Quality Assurance

An analytical process using reagent blanks and sample replication assessed the precision, bias, and pollution. The analysis showed that the bias and precision were less than 10%. The PAHs show a wide spectrum of volatility and all 16 PAHs behave the same in the chromatographic area in the standard and the sample. The set of samples was analyzed along with a blank for PAH background correction. The results were corrected for recoveries using an internal standard method, assuming that PAHs and d-PAHs behave in a similar manner during extraction and analysis. Recoveries of d-PAHs were utilized to estimate recoveries of the native PAHs. The average recoveries of d-PAHs varied from 79.11 to 91.56% with relative standard deviation (RSDs) ranging from 7.02% to 8.86%.

## 3. Results

### 3.1. Levels of PAHs in Road Dust Samples

The levels of all determined PAHs in road dust samples are shown in Table 2. Results depicted that the minimum P1 (naphthalene) concentration was 11.35 ng g^−1^, and the maximum was 52.33 ng g^−1^. The concentration of P2 (acenaphthylene) ranged from 12.31 to 46.38 ng g^−1^. The highest concentration of P3 (acenaphthene) was 43.68 ng g^−1^, and the lowest was 13.26 ng g^−1^ with an average value of 22.32 ng g^−1^. The concentration of fluorene (P4) was also determined and the maximum was 46.51 ng g^−1^ while the minimum was 12.38 ng g^−1^. The average P4 concentration among the 50 collected samples was 22.15 ng g^−1^. The highest P5 (phenanthrene) concentration was 45.92 ng g^−1^ while the lowest was 10.36 ng g^−1^ with an average of 10.36 ng g^−1^. The data showed that the highest concentration of P6 (anthracene), P7 (fluoranthene), P8 (pyrene), P9{benzo(b+j)fluoranthene}, P10{benzo(k)fluoranthene}, P11{benzo(a)pyrene}, P12 (3-methylcholanthrene), P13{dibenz(a.h)acridine}, P14{indeno(1.2.3-cd)pyrene, P15{dibenz(a.h)anthracene}, and P16{benzo(g.h.i)perylene} was 44.11 ng g^−1^, 39.60 ng g^−1^, 46.09 ng g^−1^, 38.69 ng g^−1^, 32.34 ng g^−1^, 41.55 ng g^−1^, 126.36 ng g^−1^, 33.26 ng g^−1^, 31.68 ng g^−1^, 32.61 ng g^−1^, and 41.18 ng g^−1^, respectively. The results showed that the minimum concentration of P6 (anthracene), P7 (fluoranthene), P8 (pyrene), P9 {benzo(b+j) fluoranthene}, P10 {benzo(k)fluoranthene}, P11 {benzo(a)pyrene}, P12 (3-methylcholanthrene), P13 {dibenz(a.h)acridine}, P14 {indeno(1.2.3-cd)pyrene}, P15 {dibenz(a.h)anthracene}, and P16 {benzo(g.h.i)perylene} was 13.34 ng g^−1^, 14.21 ng g^−1^, 13.08 ng g^−1^, 11.05 ng g^−1^, 12.61 ng g^−1^, 13.57 ng g^−1^, 12.33 ng g^−1^, 12.66 ng g^−1^, 10.67 ng g^−1^, 12.02 ng g^−1^, and 12.24 ng g^−1^. The concentrations of the individual PAHs in the road dust samples decreased in the order P1 > P8 > P6 > P7 > P5 > P9 > P12 > P2 > P11 > P3 > P4 > P10 > P13 > P15 > P16 > P14 on a mean basis.

### 3.2. Daily Dose Intake

Figure 2a–c presents the results of the average daily dose ingestion (ADDing). The ADDing was calculated separately for children (>6), teenagers (6–12), and adults (>12) for all PAHs with each collected sample. The highest and lowest ADDing values for P1 (naphthalene) were 6.3 × 10^−7^ mg kg^−1^ and 4.2 × 10^−7^ mg kg^−1^, respectively, for children (<6 years old) with an average value of 3.1 × 10^−7^ mg kg^−1^. The ADDing values for P2 (acenaphthylene) ranged from 3.8 × 10^−7^ mg kg^−1^ to 5.5 × 10^−7^ mg kg^−1^ while the average value was 2.7 × 10^−7^ mg kg^−1^ for children (<6 years old). The highest ADDing P3 (acenaphthene) value was 4.0 × 10^−7^ mg kg^−1^ and the lowest was 2.6 × 10^−4^ mg kg^−1^ for children (<6 years old). The results showed that the maximum ADDing of P4 (fluorene), P5 (phenanthrene), P6 (anthracene), P7 (fluoranthene), P8 (pyrene), P9 {benzo(b+j) fluoranthene}, P10 {benzo(k)fluoranthene}, P11 {benzo(a)pyrene}, P12 (3-methylcholanthrene), P13 {dibenz(a.h)acridine}, P14 {indeno(1.2.3-cd)pyrene}, P15 {dibenz(a.h)anthracene}, and P16 {benzo(g.h.i)perylene} for children (<6 years old) was 1.6 × 10^−4^ mg kg^−1^, 5.5 × 10^−4^ mg kg^−1^, 5.3 × 10^−4^ mg kg^−1^, 1.9 × 10^−4^ mg kg^−1^, 5.3 × 10^−4^ mg kg^−1^, 1.9 × 10^−4^ mg kg^−1^, 5.5 × 10^−4^ mg kg^−1^, 4.6 × 10^−7^ mg kg^−1^, 3.9 × 10^−7^ mg kg^−1^, 5.0 × 10^−7^ mg kg^−1^, 2.9 × 10^−4^ mg kg^−1^, 4.0 × 10^−7^ mg kg^−1^, 3.8 × 10^−7^ mg kg^−1^, 3.9 × 10^−6^ mg kg^−1^ and 4.9 × 10^−7^ mg kg^−1^. The ADDing in teenagers (6–12 years) and in adults (>12 years) was the following decreasing order P7 > P11 > P6 > P3 > P8 > P13 > P10 > P4 > P12 > P2 > P16 > P15 > P1 > P9>P14> P5 based on means. Children are more exposed to dust because of their physical activities, including ingesting dust through the mouth, playing and holding to sand other household objects, and licking their hands.

The average daily dose (dermal) ADDder with respect to each PAH at each sample location is shown in Figure 3. The ADDder was calculated for children, i.e., less than 6 years old, teenagers aged 6 to 12 years, and adults aged more than 12 years. The ADDder for all age individuals were in the following decreasing order P12 > P3 > P7 > P4 > P15 > P1 > P8 > P6 > P5 > P9 > P2 > P11 > P10 > P13 > P16 > P14. The highest ADDder of P1 (naphthalene), P2 (acenaphthylene), P3 (acenaphthene), P4 (fluorene), P5 (phenanthrene), P6 (anthracene), P7 (fluoranthene), P8 (pyrene), P9 {benzo(b+j) fluoranthene}, P10 {benzo(k)fluoranthene}, P11 {benzo(a)pyrene}, P12 (3-methylcholanthrene), P13 {dibenz(a.h)acridine}, P14 {indeno(1.2.3-cd)pyrene}, P15 {dibenz(a.h)anthracene}, and P16 {benzo(g.h.i)perylene} for children (<6 years old) was 4.37 × 10^−9^ mg kg^−1^, 3.87 × 10^−9^ mg kg^−1^, 1.79 × 10 mg kg^−1^, 1.15 × 10^−7^ mg kg^−1^, 3.83 × 10^−9^ mg kg^−1^, 3.68 × 10^−9^ mg kg^−1^, 1.30 × 10^−6^ mg kg^−1^, 3.85 × 10^−9^ mg kg^−1^, 3.23 × 10^−9^ mg kg^−1^, 2.70 × 10^−9^ mg kg^−1^, 3.47 × 10^−9^ mg kg^−1^, 2.04 × 10^−6^ mg kg^−1^, 2.78 × 10^−9^ mg kg^−1^, 2.64 × 10^−9^ mg kg^−1^, 2.72 × 10^−8^ mg kg^−1^, and 3.44 × 10^−9^ mg kg^−1^, respectively.

### 3.3. Lifetime Average Daily Intake of PAHs

For the human health hazard/risk calculation in this experimentation, we adopt that human adults of 70 years were exposed for all the days in a year. The estimated lifetime average daily dose (LADD) of 16 PAHs for human adults is shown in Figure 4. The LADD with respect to individual PAHs in the road dust samples decreased in the order P3 > P15 > P1 > P8 > P6 > P7 > P5 > P9 > P2 > P12 > P11 > P4 > P10 > P13 > P16 > P14. The LADD with respect to naphthalene (P1) ranged from 1135 mg kg^−1^ day^−1^ to 5233 mg kg^−1^ day^−1^ with an average value of 2556 mg kg^−1^ day^−1^ among 50 collected samples of road dust in Riyadh City, Saudi Arabia. The highest LADD with respect to acenaphthylene (P2) was found with the value 4638 mg kg^−1^ day^−1^ and the lowest was 1231 mg kg^−1^ day^−1^ in collected samples. The results displayed that the maximum LADD of P3 (acenaphthene), P4 (fluorene), P5 (phenanthrene), P6 (anthracene), P7 (fluoranthene), P8 (pyrene), P9 {benzo(b+j) fluoranthene}, P10 {benzo(k)fluoranthene}, P11 {benzo(a)pyrene}, P12 (3-methylcholanthrene), P13 {dibenz(a.h)acridine}, P14 {indeno(1.2.3-cd)pyrene}, P15 {dibenz(a.h)anthracene}, and P16 {benzo(g.h.i)perylene} was 31,886 mg kg^−1^ day^−1^, 4651 mg kg^−1^ day^−1^, 4592 mg kg^−1^ day^−1^, 4411 mg kg^−1^ day^−1^, 3960 mg kg^−1^ day^−1^, 4609 mg kg^−1^ day^−1^, 3869 mg kg^−1^ day^−1^, 3234 mg kg^−1^ day^−1^, 4155 mg kg^−1^ day^−1^, 12,636 mg kg^−1^ day^−1^, 3326 mg kg^−1^ day^−1^, 3168 mg kg^−1^ day^−1^, 32,611 mg kg^−1^ day^−1^, and 4118 mg kg^−1^ day^−1^, respectively. While the lower LADD values with respect to of P1 (naphthalene), P2 (acenaphthylene), P3 (acenaphthene), P4 (fluorene), P5 (phenanthrene), P6 (anthracene), P7 (fluoranthene), P8 (pyrene), P9 {benzo(b+j) fluoranthene}, P10 {benzo(k)fluoranthene}, P11 {benzo(a)pyrene}, P12 (3-methylcholanthrene), P13 {dibenz(a.h)acridine}, P14 {indeno(1.2.3-cd)pyrene}, P15 {dibenz(a.h)anthracene}, and P16 {benzo(g.h.i)perylene} was 1135 mg kg^−1^ day^−1^, 1231 mg kg^−1^ day^−1^, 1562 mg kg^−1^ day^−1^, 1238 mg kg^−1^ day^−1^, 1036 mg kg^−1^ day^−1^, 1334 mg kg^−1^ day^−1^, 1421 mg kg^−1^ day^−1^, 1308 mgkg^−1^ day^−1^, 1105 mg kg^−1^ day^−1^, 1261 mg kg^−1^ day^−1^, 1357 mg kg^−1^ day^−1^, 1233 mg kg^−1^ day^−1^, 1266 mg kg^−1^ day^−1^, 1067 mg kg^−1^ day^−1^, 1202 mg kg^−1^ day^−1^, and 1224 mg kg^−1^ day^−1^, respectively.

### 3.4. Toxicity Equivalent Factor

The BaP has enough toxicology evidence to use as the basis for a toxicity factor. As a result, BaP toxic equivalent factors were used to measure the volume of toxicity in Riyadh City road dust, as suggested for our specific PAHs compounds. The benzo[a]pyrene toxicity equivalency (BaPTEQ) represents the carcinogenic ability of PAHs that conform to BaP. The TEF data were adapted from the TEQBaP according to Nisbet and Lagoy [36]. Results are presented in Table 3 along with the minimum, maximum, and mean concentration of each PAH investigated in the study. The largest TEQBaP was detected with P15 dibenz(a.h)anthracene, whereas the lowest was found with the same value in P2 (acenaphthene), P4 (phenanthrene), P5 (phenanthrene), and P7 (fluoranthene). BaP was used as a basis for determining TEF, and values were allocated based on [36] descriptions. Other PAHs were given TEF values based on their carcinogenicity in comparison to BaP.

## 4. Discussion

The concentration of PAHs in dust varies with the nature of sampling areas such as rural and urban. Urban areas have dense road structures with heavy traffic, automobiles, hospitals, industrial sites, railways, waste dumping sites, etc. This study was purely focused on an urban city, i.e., Riyadh. Our results are similar to previous studies in terms of levels and occurrence of PAHs in road dust samples [37]. The total PAHs (ΣPAHs) in urban street dust from Tianjin ranged from 538 μg kg^−1^ to 34.3 mg kg^−1^, with a mean value of 7.99 mg kg^−1^. As compared to this study, overall, the concentration of ∑PAHs was from 0.01 to 126 ng g^−1^ (Table 2). [18] depicted that acenaphthene (PAH) varied in summer and winter ranging from 3 ng g^−1^ to 1485 ng g^−1^ and 50 ng g^−1^ to 3162 ng g^−1^, respectively. Organic contamination in road dust was rich in PAHs that have more than 4 benzene rings and a high molecular weight was present, confirming the 4 and above benzene rings. [38] determined various PAHs concentrations in different areas of Bayreuth, Germany. Several sites with heavy industries, including gas plants and three railroad areas, had a large concentration of different PAHs (the sum of 20 PAHs). The concentration of these PAHs was similar to our findings ranging from 2.40 to 48.90 ng g^−1^ soil. The levels in the center of the city ranged from 0.63 to 20.7 ng g^−1^ soil.

The World Health Organization (WHO) and their agency namely International Agency for Research on Cancer [39] defined 7 PAHs as possible carcinogens for humans, i.e., indeno(1,2,3,cd)pyrene, dibenzo(a,h)anthracene, benzo(k)fluoranthene, chrysene, benzo(a)anthracene, benzo(b)fluoranthene and BaP. The concentrations of 7PAH which are potential carcinogens observed in soils from Gwalior, India, ranged between 41–460 ng g^−1^ with a mean of 181 ± 51 ng g^−1^ and accounted for 38% of 16 PAHs. The average concentration of individual probable human carcinogens was 27 ± 5 ng g^−1^ {forbenzo(a)athracene}, 71 ± 28 ng g^−1^ (chrysene), 22 ± 6 ng g^−1^ {benzo(b)fluoranthene}, 14 ± 4 ng g^−1^ {benzo(k)fluoranthene}, 17 ± 3 ng g^−1^ (BaP), 25 ± 3 ng g^−1^ {dibenzo(a,h)anthracene}, and 22 ± 3 ng g^−1^ {indeno(1,2,3-cd)pyrene}, respectively. The combinations of the benzene ring in PAHs revealed different properties concerning carcinogen nature and exposure to humans. The release of PAHs into the environment may change due to pyrogenic and petrogenic sources as these are the most important factors [40]. It has been depicted that previously, PAHs, which possess two to three rings and are considered low molecular weight, arise from petrogenic sources, while PAHs having four to six rings and have high molecular weight arise from pyrogenic sources such as natural gas, fossil fuels, diesel, coal, and combustion of gasoline [41,42]. Typically, low-molecular-weight PAHs with 2 to 4 benzene rings are in the category of non-carcinogenic but studies showed their toxicity towards aquatic organisms, while the high-molecular-weight PAHs with 5 and 6 member-ring PAHs are classified in the category of carcinogenic [43]. The PAHs with five and six rings were predominant in suspended sediments. Our results are similar to previous studies, which stated that PAHs are the main contributor to settled dust [44,45]. Our results indicated that dust is an important source of 3–5 ring PAHs, but we need to be cautious since the collected samples and dataset in this experiment were minor and need more detailed studies to authenticate this point.

The LADD is the amount of intake per kg of body weight per day of a chemical (e.g., PAHs) suspected of having adverse health effects when absorbed into the body over a long period. In this study, we also determined that residents in the age-specific group are exposed for 350 days a year during their life span. The estimated LADD values of individual PAH for an age-specific group are shown in Figure 4. Our results are in agreement with [46,47]. The ADDing showed a real picture of PAHs contamination in dust concerning human health. The ADDing was higher in children as compared to adults. Our results are in agreement with [48]. Moreover, children can receive 2.5-fold more carcinogenic PAHs through dust ingestion as compared to inhalation [49,50]. As children have lower body weight, the intake of PAHs (mg per kg of body weight per day) is believed to be greater as compared to adults. Children are also undergoing early organ, immune system, and nervous development; hence, they are becoming more sensitive to carcinogens [51]. Owing to such factors, children are at more risk of PAHs as compared to adults. (Hu et al. (2007) [52] revealed that urban dust of Tianjing (China) created human cancer risks because of exposure to PAHs because of higher values of dermal exposure (SA) along with the increased value of exposure duration in Children. While results are similar to) [53] Yang et al. (2015), who researched the soil samples of Guizhou province, southwest China, and identified two major sources of PAH comprising vehicle emissions and coal combustion. High molecular weight and TEQBaP in Riyadh dust suggest that high molecular weight species have much higher TEF values than low molecular weight species. Also measured the TEQ of various PAHs in street dust of Bushehr City, Iran. They found a positive correlation between high molecular weight PAHs and TEQ. The PAHs released from fuel combustion, have higher TEF than low molecular weight PAHs and mostly affect TEQBaP same as in the current study. Alghamdi et al. (2022) [34] (reported that the rapid industrialization and urbanization and their concentrations in dust may cause health problems in the near future in the north side as well as other sides of Riyadh City.

## 5. Conclusions

This study presents PAH concentration in the road dust of Riyadh City, Saudi Arabia. The 16 PAHs were determined from 50 different locations. The findings revealed variable concentration levels for all tested PAHs. Many factors altogether impacted the PAH concentrations in road dust samples such as traffic density, vehicle emissions, industrial emissions, pollutant accumulation/road dust, and sampling site locations. The LADD showed high values. The ADDing and ADDderm depicted more values for children as compared to adults. The total equivalency factor based on BaP (TEQBaP) calculations revealed that PAHs with more benzene rings or having high molecular weight showed high TEQBaP as compared to low molecular weight PAHs. In the future, there is a great need for the identification of spatial-temporal control drivers in Saudi Arabia that determines the dust storms events. Furthermore, there is a need to emphasize significant correlations between human health problems and road dust, particularly in densely populated areas.

## Figures and Tables

**Figure 1 ijerph-20-01216-f001:**
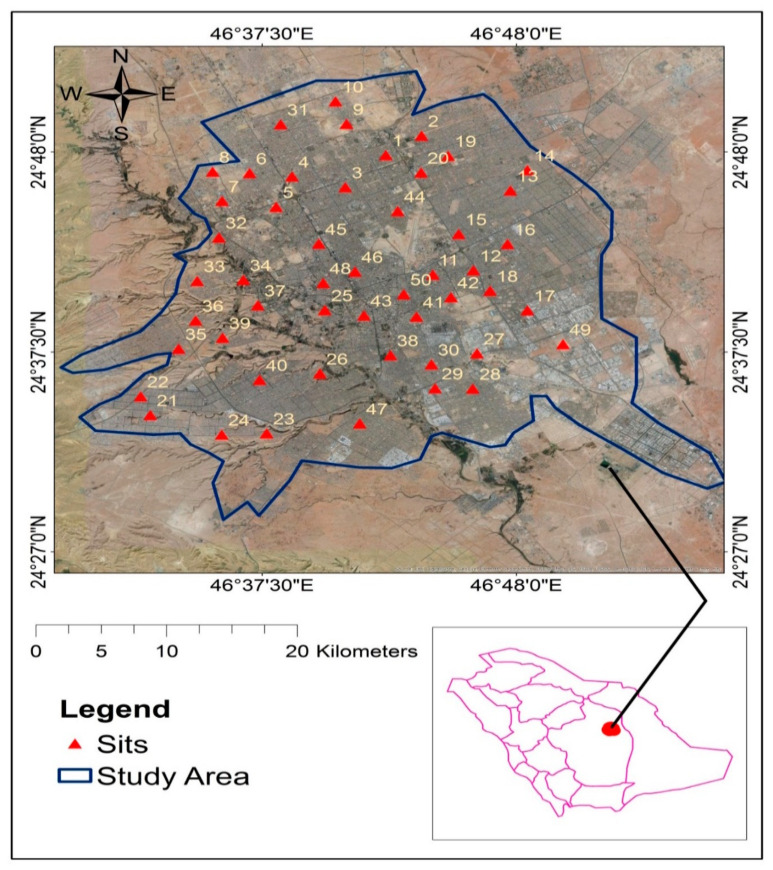
Sampling locations of air dust samples in Riyadh City, Saudi Arabia.

**Figure 2 ijerph-20-01216-f002:**
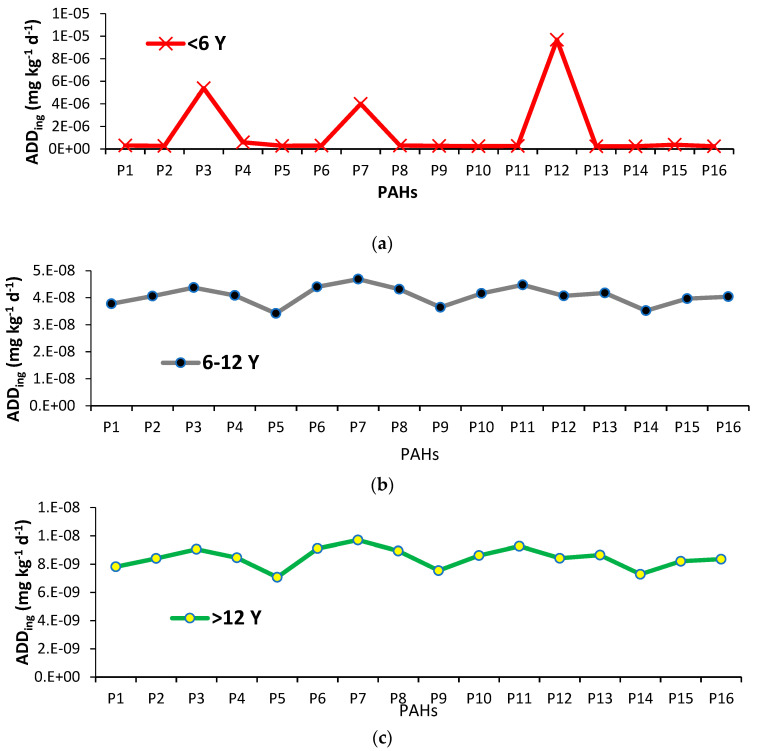
(**a**) Exposure to human body hand-to-mouth ingestion absorption against each PAH for children (<6 years) (**b**) teenagers and (**c**) adults based on mean values P1: naphthalene, P2: acenaphthylene, P3: acenaphthene, P4: fluorene, P5: phenanthrene, P6: anthracene, P7: fluoranthene, P8: pyrene, P9: benzo(b+j)fluoranthene, P10: benzo(k)fluoranthene, P11: benzo(a)pyrene, P12: 3-methylcholanthrene, P13: dibenz(a.h)acridine, P14: indeno(1.2.3-cd)pyrene, P15: dibenz(a.h)anthracene, P16: benzo(g.h.i)perylene.

**Figure 3 ijerph-20-01216-f003:**
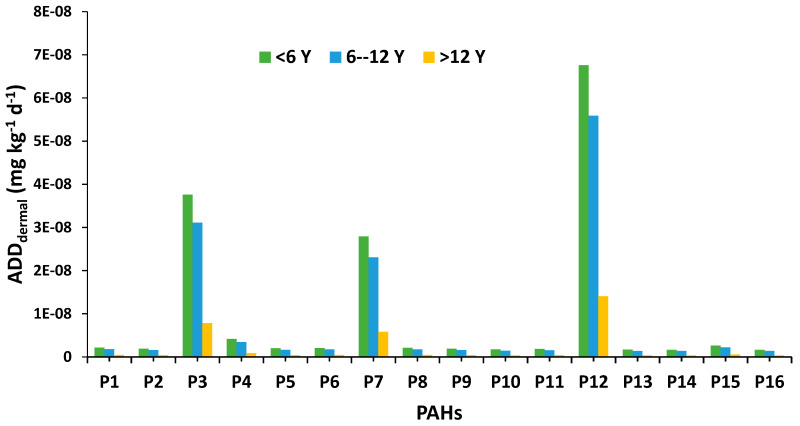
Exposure to human body hand-to-mouth dermal absorption for each PAH. P1: naphthalene, P2: acenaphthylene, P3: acenaphthene, P4: fluorene, P5: phenanthrene, P6: anthracene, P7: fluoranthene, P8: pyrene, P9: benzo(b+j)fluoranthene, P10: benzo(k)fluoranthene, P11: benzo(a)pyrene, P12: 3-methylcholanthrene, P13: dibenz(a.h)acridine, P14: indeno(1.2.3-cd)pyrene, P15: dibenz(a.h)anthracene, P16: benzo(g.h.i)perylene.

**Figure 4 ijerph-20-01216-f004:**
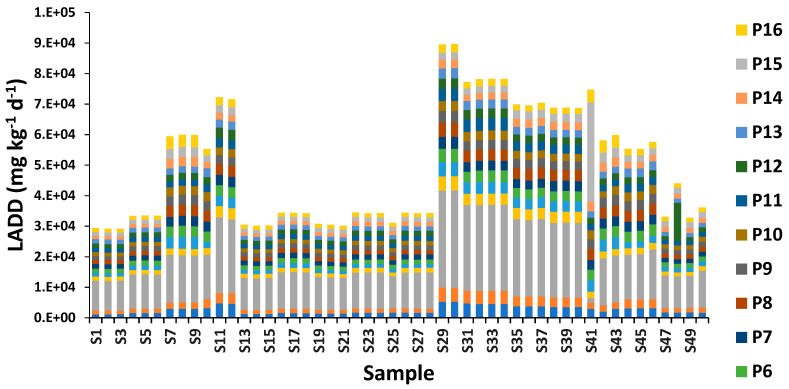
The lifetime average daily dose (LADD) of sixteen PAH against sampling locations. P1: naphthalene, P2: acenaphthylene, P3: acenaphthene, P4: fluorene, P5: phenanthrene, P6: anthracene, P7: fluoranthene, P8: pyrene, P9: benzo(b+j)fluoranthene, P10: benzo(k)fluoranthene, P11: benzo(a)pyrene, P12: 3-methylcholanthrene, P13: dibenz(a.h)acridine, P14: indeno(1.2.3-cd)pyrene, P15: dibenz(a.h)anthracene, P16: benzo(g.h.i)perylene.

**Table 1 ijerph-20-01216-t001:** Parameter meaning and value of daily dose model of PAHs metals in urban surface dusts.

Factors	Meaning and Unit	Values	
		Child	Adult
IngR	Ingestion rate, mg/d	200	100
InhR	Inhalation rate, m^3^/d	7.6	20
PEF particle	Emission factor, m^3^/kg	1.36 × 109	
SA exposure	Skin area, cm^2^	2800	5700
SL skin	Adherence factor, g/(cm^2^h)	0.2	0.7
ABS	Dermal absorption factor, (unitless)	0.001	
ED	Exposure duration, d/y	6	24
EF	Exposure frequency, d/y	180	
BW	Average body weight, kg	15	70
AT	Average time, ds	ED × 365 (for non-carcinogens) 70 × 365 (for carcinogens)	
C	Exposure-point concentration, mg/kg	95% UCL	

Sources: [16,34].

**Table 2 ijerph-20-01216-t002:** Concentration of PAHs in road dust samples of Riyadh City.

**Sample**	**P1**	**P2**	**P3**	**P4**	**P5**	**P6**	**P7**	**P8**	**P9**	**P10**	**P11**	**P12**	**P13**	**P14**	**P15**	**P16**
	**(ng g^−1^)**															
**S1**	11.35	12.57	13.38	13.83	11.84	14.09	15.53	13.55	11.54	12.97	14.33	13.77	14.59	11.82	12.02	12.48
**S2**	11.45	12.44	13.26	1,377	11.79	14.07	15.31	13.52	11.57	12.62	14.54	13.69	14.66	11.15	12.16	12.24
**S3**	11.48	12.31	13.38	13.74	11.54	14.04	15.41	13.5	11.05	12.72	14.31	13.48	14.39	11.31	12.23	12.38
**S4**	15.68	14.35	15.32	14.35	14.61	16.43	16.22	14.44	16.25	13.65	16.24	15.05	15.78	12.37	13.57	12.98
**S5**	15.64	14.38	15.45	14.29	14.26	16.42	16.73	14.36	16.21	13.61	16.22	15.07	15.78	12.44	13.61	12.92
**S6**	15.66	14.41	15.36	14.3	14.16	16.45	16.78	14.48	16.21	13.61	16.36	15.03	15.71	12.41	13.49	12.97
**S7**	29.21	20.22	21.42	20.81	37.27	34.71	31.6	36.34	31.11	29.26	20.22	21.42	20.81	31.08	32.11	41.18
**S8**	29.31	21.28	21.09	21.81	39.26	35.78	32.61	36.32	33.14	29.03	21.28	21.04	21.82	30.8	31.79	40.11
**S9**	29.21	21.22	21.04	20.85	39.23	35.71	32.68	36.53	33.19	29.21	21.52	21.38	20.86	31.68	31.35	40.28
**S10**	31.52	29.18	20.01	21.08	20.29	34.15	32.81	35.57	31.49	29.26	26.71	22.67	29.27	21.22	21.04	20.86
**S11**	46.89	34.49	33.95	36.41	34.09	34.83	33.94	36.36	27.42	26.11	29.92	34.39	26.61	22.51	24.29	26.38
**S12**	46.03	34.22	33.22	36.48	34.14	34.75	33.84	36.28	27.41	26.39	29.62	34.31	26.62	22.53	24.38	26.39
**S13**	12.33	13.69	14.39	13.83	12.81	14.51	15.53	13.08	13.49	12.94	14. 73	13. 23	12. 92	10. 86	12. 02	13. 86
**S14**	12.38	13.45	14.21	13.39	12.64	14.26	15.2	13.13	13.58	12.61	14.57	13.08	12.66	11.16	12.13	13.24
**S15**	12.72	13.39	14.38	13.46	12.11	14.35	15.11	13.24	13.05	12.73	14.39	13.42	12.89	10.67	12.23	13.29
**S16**	15.58	14.59	16.33	14.39	15.67	16.6	16.7	14.78	16.12	13.72	16.12	15.56	15.77	12.89	13.81	12.55
**S17**	15.24	14.38	16.44	14.26	15.23	16.47	17.73	14.38	16.27	13.89	16.23	15.50	15.11	12.66	13.7	12.59
**S18**	15.11	14.45	16.36	14.32	15.13	16.43	16.78	14.67	16.29	13.22	16.36	15.59	15.32	12.82	13.58	12.54
**S19**	13.44	12.98	14.77	12.59	10.88	13.61	14.33	13.88	12.84	13.84	14.22	14.59	13	12.84	13.61	12.77
**S20**	13.51	12.67	14.55	12.66	11.15	13.44	14.21	13.37	12.63	13.82	14.21	14.26	13.12	12.63	13.44	12.67
**S21**	13.71	12.7	14.37	12.38	10.36	13.34	14.36	13.44	12.11	13.76	14.77	14.34	13.29	12.11	13.34	12.46
**S22**	16.74	13.73	16.13	15.77	12.89	14.58	16.32	14.36	15.66	14.88	16.62	16.68	14.77	15.66	14.58	13.53
**S23**	16.25	13.89	16.23	15.11	12.66	14.38	16.44	14.26	15.23	14.77	16.61	16.63	14.38	15.23	14.38	13.44
**S24**	16.24	13.22	16.36	15.32	12.82	14.45	16.36	14.32	15.13	14.56	16.35	16.66	14.67	15.13	14.45	13.67
**S25**	16.75	14.7	14.37	12.38	12.36	13.34	14.36	13.44	12.11	13.76	14.77	14.34	13.29	14.11	13.34	12.46
**S26**	16.74	13.73	16.13	15.77	12.89	14.58	16.32	14.36	15.66	14.88	16.62	16.68	14.77	15.66	14.58	13.00
**S27**	16.29	13.89	16.23	15.11	12.66	14.38	16.44	14.26	15.23	14.77	16.61	16.63	14.38	15.23	14.38	13.44
**S28**	16.23	13.22	16.36	15.32	12.82	14.45	16.36	14.32	15.13	14.56	16.35	16.66	14.67	15.13	14.45	13.66
**S29**	52.33	46.38	43.59	46.51	45.92	44.11	39.6	46.09	38.64	31.34	40.24	34.1	33.26	29.29	22.63	26.83
**S30**	52.06	46.24	43.68	46.5	45.75	44.1	39.6	46.03	38.69	32.34	40.3	34.19	33.26	29.42	22.49	26.85
**S31**	47.33	41.19	38.52	36.51	37.27	34.71	31.6	36.34	31.6	31.34	40.24	31.11	29.26	20.22	21.42	20.81
**S32**	45.36	43.2	38.52	36.51	38.2	37.76	31.77	36.88	30.62	31.00	41.24	31.19	29.47	22.29	21.88	22.81
**S33**	45.34	43.13	38.62	36.58	38.27	37.82	31.63	36.91	30.69	31.3	41.29	31.11	29.46	22.67	21.78	22.81
**S34**	45.3	43.19	38.6	36.55	38.2	37.69	31.61	36.72	30.77	31.1	41.55	31.13	29.56	22.91	21.82	22.86
**S35**	37.16	33.15	34.59	36.68	34.59	31.19	29.18	36.75	30.36	25.38	29.18	24.39	25.35	26.3	26.89	20.02
**S36**	37.16	33.16	34.15	35.57	34.15	31.11	29.26	36.74	30.74	25.39	29.26	24.66	25.88	26.44	26.83	20.22
**S37**	37.16	33.19	34.71	36.34	34.71	31.13	29.47	36.8	30.8	25.55	29.47	24.77	25.35	26.77	26.8	22.29
**S38**	35.16	31.37	33.47	36.46	34.03	32.93	33.73	36.38	28.46	24.38	29.35	24.44	24.33	26.32	26.68	19.58
**S39**	35.03	31.44	33.6	36.46	34.18	32.82	33.93	36.36	28.42	24	29.28	24.37	24.29	26.3	26.61	19.51
**S40**	35.27	31.22	33.43	36.46	34.12	32.71	33.83	36.27	28.44	24.32	29.29	24,391	24.36	26.39	26.60	19.50
**S41**	29.26	20.22	21,423	20.81	37.27	34.71	31.6	36.34	31.11	29.26	20.22	21.42	20.81	31.08	326.11	41.18
**S42**	20.03	21.28	21.09	21.81	39.26	35.78	32.61	36.32	33.14	20.03	21.28	21.09	21.81	30.81	31.79	40.11
**S43**	29.27	21.22	21.04	20.86	39.27	35.71	32.6	36.52	33.11	29.27	21.22	21.04	20.86	31.6	31.34	40.24
**S44**	31.02	29.18	20.02	21.08	20.21	34.15	32.81	35.57	31.49	29.2	26.71	22.67	29.27	21.22	21.04	20.86
**S45**	31.11	29.26	20.22	21.42	20.81	34.71	31.6	36.34	33.18	28.02	26.08	20.22	29.33	21.26	21.4	20.83
**S46**	31.19	29.47	22.29	21.88	22.81	37.76	31.77	36.88	31.26	29.22	26.42	21.81	29.36	21.34	21.4	20.83
**S47**	17.67	14.55	14.56	13.55	14.83	14.8	14.61	14.55	15.52	14.33	13.99	12.66	13.89	16.89	18.44	14.22
**S48**	17.05	14.21	14.28	13.18	14.22	14.62	14.44	14.66	15.21	14.82	13.59	126.36	13.81	16.67	18.66	14.66
**S49**	17.68	14.74	14.37	13.29	14.11	14.44	14.34	14.41	15.76	14.55	13.57	12.33	13.77	16.46	18.69	14.63
**S50**	16.61	16.98	16.6	14.7	15.63	15.57	15.58	16.17	16.67	16.62	14.66	15.81	14.33	16.05	19.45	15.22

P1: naphthalene, P2: acenaphthylene, P3: acenaphthene, P4: fluorene, P5: phenanthrene, P6: anthracene, P7: fluoranthene, P8: pyrene, P9: benzo(b+j)fluoranthene, P10: benzo(k)fluoranthene, P11: benzo(a)pyrene, P12: 3-methylcholanthrene, P13: dibenz(a.h)acridine, P14: indeno(1.2.3-cd)pyrene, P15: dibenz(a.h)anthracene, P16: benzo(g.h.i)perylene.

**Table 3 ijerph-20-01216-t003:** Toxic equivalency concentration of 16 PAHs based on BaPeq.

PAH	Min	Max	Mean	TEF *	TEQ_BaP_
P1	11.35 ng g^−1^	52.330 ng g^−1^	46.890 ng g^−1^	0.001	0.05
P2	12.31 ng g^−1^	46.38 ng g^−1^	22.55 ng g^−1^	0.001	0.02
P3	13.26 ng g^−1^	2142 ng g^−1^	450.35 ng g^−1^	0.001	0.45
P4	12.38 ng g^−1^	46.51 ng g^−1^	22.15 ng g^−1^	0.001	0.02
P5	10.36 ng g^−1^	45.92 ng g^−1^	23.51 ng g^−1^	0.001	0.02
P6	13.34 ng g^−1^	44.11 ng g^−1^	24.70 ng g^−1^	0.01	0.25
P7	14.21 ng g^−1^	39.60 ng g^−1^	23.86 ng g^−1^	0.001	0.02
P8	13.08 ng g^−1^	46.09 ng g^−1^	25.21 ng g^−1^	0.001	0.03
P9	11.05 ng g^−1^	38.69 ng g^−1^	22.64 ng g^−1^	0.1	2.26
P10	12.61 ng g^−1^	32.34 ng g^−1^	20.68 ng g^−1^	0.1	2.07
P11	13.57 ng g^−1^	41.55 ng g^−1^	22.34 ng g^−1^	1	22.34
P12	12.33 ng g^−1^	2439 ng g^−1^	519.87 ng g^−1^	0.001	0.52
P13	12.66 ng g^−1^	33.26 ng g^−1^	20.21 ng g^−1^	5	101.03
P14	10.67 ng g^−1^	31.68 ng g^−1^	19.67 ng g^−1^	0.1	1.97
P15	12.02 ng g^−1^	32.61 ng g^−1^	25.90 ng g^−1^	5	129.52
P16	12.24 ng g^−1^	41.18 ng g^−1^	19.86 ng g^−1^	0.01	0.20

P1: naphthalene, P2: acenaphthylene, P3: acenaphthene, P4: fluorene, P5: phenanthrene, P6: anthracene, P7: fluoranthene, P8: pyrene, P9: benzo(b+j)fluoranthene, P10: benzo(k)fluoranthene, P11: benzo(a)pyrene, P12: 3-methylcholanthrene, P13: dibenz(a.h)acridine, P14: indeno(1.2.3-cd)pyrene, P15: dibenz(a.h)anthracene, P16: benzo(g.h.i)perylene. * Toxicity equivalency factors (TEFs) for individual PAH based on [36].

## Data Availability

Not applicable.

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
