# Peer review of "Impact of Atmospheric Polycyclic Aromatic Hydrocarbons (PAHs) of Falling Dust in Urban Area Settings: Status, Chemical Composition, Sources and Potential Human Health Risks"

_ijerph, 2023, doi:10.3390/ijerph20021216_

Round 1

Reviewer 1 Report

In this study, authors measured the concentrations of 16 PAHs at 50 residential sites in Riyadh city in Saudi Arabia and further estimated the associated risk to human health. The research topic is interesting and important to research field, but the current version of manuscript lacks several necessary information in methodology and in-depth analysis in discussion. My other main comments are as follows:

1. Abstract: there is no need to show the whole rankings of concentrations of 16 PAHs and put more words on in-depth results, for example, the spatial distribution of PAHs concentrations, the health effects and even the policy suggestions based on the results.

2. Introduction: the current structure is not clear, the second half of the second paragraph is talking about road dust but in the first half of the third one is still on it, I highly suggest summarizing the results from literature and highlighted the aims and research design of this study.

3. Section 2.1 and 2.2: what is the sampling period/seasons? Was each sample measured in same time? What is the criterial for selecting the “representative samples”? All these information is very important because PAHs are from local emissions and varied from season to season.

4. Table 2: the current format looks weird, particularly for the Reference column.

5. Line 175: Please add the reference for “Corresponding TEF”.

6. Line 215: There is no table legend, and the typesetting is hard for reading.

7. section 3.2: I don’t understand how authors can obtain results for different age groups. Can authors point out the description about population groups for different ages in the manuscript?

Author Response

Ans: English improved by native speaker

Comments and Suggestions for Authors

In this study, authors measured the concentrations of 16 PAHs at 50 residential sites in Riyadh city in Saudi Arabia and further estimated the associated risk to human health. The research topic is interesting and important to research field, but the current version of manuscript lacks several necessary information in methodology and in-depth analysis in discussion. My other main comments are as follows:

  1. Abstract: there is no need to show the whole rankings of concentrations of 16 PAHs and put more words on in-depth results, for example, the spatial distribution of PAHs concentrations, the health effects and even the policy suggestions based on the results.

Ans: The abstract revised accordingly and health effects and even the policy suggestions based on the results added at the end.

  1. Introduction: the current structure is not clear, the second half of the second paragraph is talking about road dust but in the first half of the third one is still on it, I highly suggest summarizing the results from literature and highlighted the aims and research design of this study.

Ans. The introduction part revised as per suggestions.

  1. Section 2.1 and 2.2: what is the sampling period/seasons? Was each sample measured in same time? What is the criterial for selecting the “representative samples”? All these information is very important because PAHs are from local emissions and varied from season to season.

Ans. The season, criteria of sampling added in respective place of methodology section.

  1. Table 2: the current format looks weird, particularly for the Reference column.

Ans:  Table 2 revised and references shifted to appropriate place.

  1. Line 175: Please add the reference for “Corresponding TEF”.

Ans: Needful done

  1. Line 215: There is no table legend, and the typesetting is hard for reading.

Ans: Table legend added (Table 2)

  1. section 3.2: I don’t understand how authors can obtain results for different age groups. Can authors point out the description about population groups for different ages in the manuscript?

Ans: The data was obtained by adopting factors developed by Zheng et al. 2010; Kaur et al. 2022. The description added in the results.

Reviewer 2 Report

The authors measured 16 PAHs in 50 locations of Riyadh city, Saudi Arabia to evaluate their sources, concentration, and compositions. The authors ranked the concentration of PAHs in the road dust samples and found the exposures were more in children than teenagers and adults. This is an interesting topic and fits the readership of IJERPH. The manuscript is well written. I recommend publication after addressing the following comments.

Specific comments:

Table 1, the table is very hard for readers to read and visualize. Consider changing it to a bar chart or just show the most significant results and move the rest to the SI.

Section 3.2, are the different ADD for different age groups caused by different constants in the calculation? Which species has the largest impact?

Author Response

Comments and Suggestions for Authors

The authors measured 16 PAHs in 50 locations of Riyadh city, Saudi Arabia to evaluate their sources, concentration, and compositions. The authors ranked the concentration of PAHs in the road dust samples and found the exposures were more in children than teenagers and adults. This is an interesting topic and fits the readership of IJERPH. The manuscript is well written. I recommend publication after addressing the following comments.

Specific comments:

Table 1, the table is very hard for readers to read and visualize. Consider changing it to a bar chart or just show the most significant results and move the rest to the SI.

 Ans: Table shifted to SI.

Section 3.2, are the different ADD for different age groups caused by different constants in the calculation? Which species has the largest impact?

Ans: As described in results and discussion the Children population is at high risk for asthma, respiratory and cardiovascular diseases, and immunity suppression as compared to adults in the particular area of investigated region

Round 2

Reviewer 1 Report

The revised version improved a lot. I believe it can be accepted after double checking all tables and figures. For example, some contents can’t be seen in Table 2 and Figure 2 missed the label “a” for the first subfigure.

Author Response

The revised version improved a lot. I believe it can be accepted after double checking all tables and figures. For example, some contents can’t be seen in Table 2 and Figure 2 missed the label “a” for the first subfigure.

Authors Reply

Reviewer 1 Comments

Corrected in R2 Manuscript

some contents can’t be seen in Table 2

Added and Corrected in R2 Manuscript

Figure 2 missed the label “a” for the first subfigure.